# Cost-effectiveness of single-layer versus double-layer uterine closure during caesarean section on postmenstrual spotting: economic evaluation alongside a randomised controlled trial

Sanne I. Stegwee ,[1] Ângela J. Ben ,[2] Mohamed El Alili ,[2]
Lucet F. van der Voet ,[3] Christianne J.M. de Groot ,[1] Judith E. Bosmans ,[2]
Judith A.F. Huirne ,[1] for the 2Close study group

► Prepublication history and additional online supplemental material for this paper are available online. To view these files, please visit the journal online (http://dx.doi.org/10.1136/bmjopen-2020-044340).

**Correspondence to**
Sanne I. Stegwee;
s.stegwee@amsterdamumc.nl and
Professor Judith A.F. Huirne;
j.huirne@amsterdamumc.nl

## ABSTRACT

**Objective** To evaluate the cost-effectiveness of double-layer compared with single-layer uterine closure after a first caesarean section (CS) from a societal and healthcare perspective.

**Design** Economic evaluation alongside a multicentre, double-blind, randomised controlled trial.

**Setting** 32 hospitals in the Netherlands, 2016–2018.

**Participants** 2292 women ≥18 years undergoing a first CS were randomly assigned (1:1). Exclusion criteria were: inability for counselling, previous uterine surgery, known menstrual disorder, placenta increta or percreta, pregnant with three or more fetuses. 1144 women were assigned to single-layer and 1148 to double-layer closure. We included 1620 women with a menstrual cycle in the main analysis.

**Interventions** Single-layer unlocked uterine closure and double-layer unlocked uterine closure with the second layer imbricating the first.

**Main outcome measures** Spotting days, quality-adjusted life-years (QALYs), and societal costs at 9 months of follow-up. Missing data were imputed using multiple imputation.

**Results** No significant differences were found between single-layer versus double-layer closure in mean spotting days (1.44 and 1.39 days; mean difference (md) −0.056, 95% CI −0.374 to 0.263), QALYs (0.663 and 0.658; md −0.005, 95% CI −0.015 to 0.005), total healthcare costs (€744 and €727; md €−17, 95% CI −273 to 143), and total societal costs (€5689 and €5927; md €238, 95% CI −624 to 1108). The probability of the intervention being cost-effective at willingness-to-pay of €0, €10 000 and €20 000/QALY gained was 0.30, 0.27 and 0.25, respectively, (societal perspective), and 0.55, 0.41 and 0.32, respectively, (healthcare perspective).

**Conclusion** Double-layer uterine closure is not cost-effective compared with single-layer uterine closure from both perspectives. If this is confirmed by our long-term reproductive follow-up, we suggest to adjust uterine closure technique guidelines.

**Trial registration number** NTR5480/NL5380.

### Strengths and limitations of this study

► This study has been performed alongside a large multicentre randomised controlled trial, which is considered the best vehicle for economic evaluations.
► We prospectively collected data regarding costs and effect, and we used patient level information.
► All relevant costs for decision making were included in the analysis, to conduct an analysis from a societal perspective.
► Possible recall bias due to retrospective self-reported questionnaire over 3-month and 6-month period.
► Generalisability of the results to other populations may be limited, as other healthcare practices and payment systems may exist.

## INTRODUCTION

Caesarean section (CS) rates rise globally and is the mode of delivery for approximately one in five live births globally.[1 2] As a consequence, a rise in morbidity related to CS is observed as well.[3] Severe morbidity associated with a subsequent pregnancy includes caesarean scar pregnancy, placenta accreta spectrum disorders and uterine rupture. However, less severe but more prevalent gynaecological morbidity related to a CS have recently gained more interest as well. Chronic maternal morbidity after CS includes dysmenorrhoea and abnormal uterine bleeding, which are both associated with a sonographically visible indentation at the site of the previous uterine incision.[4–6] This indentation is called a niche and is seen in approximately 60% of women after CS.[7 8] Of them, 30% develops abnormal uterine bleeding and more specifically, postmenstrual spotting.[5 6] This is brownish

discharge at the end of the menstruation or blood loss in between two menstruations that is limiting women in daily life.[5] Over the last years, an increase in the development of medical treatments and surgical procedures to treat or remove the niche is observed, primarily aiming to reduce spotting.[9]

CS is the most common major surgical intervention.[10] However, there is no international guideline on the most optimal way to close the uterine incision while the specific closure technique may influence healing of the uterine wound. A specific issue on which no consensus exists is whether to use single-layer or double-layer closure of the uterine layers.[11 12] When comparing these techniques, no differences were found at short-term except for longer operation time after double-layer closure.[13 14] Nevertheless, previous studies also suggested that double-layer closure may result in better uterine scar healing and lower prevalence of large niches thereby possibly leading to lower medical costs than single-layer closure.[13 15] There is, however, a lack of studies on uterine closure techniques and their impact on maternal health outcomes related to gynaecological symptoms.[14] The impact of different uterine closure techniques on healthcare utilisation, informal care and lost productivity costs has never been investigated previously. As decision-makers increasingly demand evidence of cost-effectiveness (CE) of healthcare interventions, conducting economic analysis alongside clinical trials is desirable because it allows the prospective collection of cost and effect data and the use of patient level information for drawing inferences about additional costs and benefits of interventions.[16] In addition, regulatory and reimbursement agencies of many countries consider evidence of economic value along with clinical effectiveness.[16]

Therefore, the aim of this study was to perform a CE analysis of double-layer compared with single-layer uterine closure after a first CS from both a societal and healthcare perspective. We hypothesised that double-layer closure would reduce postmenstrual spotting and total societal and healthcare costs compared with single-layer closure as a result of less morbidity, despite slightly higher intervention costs of double-layer closure.

## METHODS
### Study design
An economic evaluation was performed alongside a multi-centre randomised controlled superiority trial comparing double-layer closure and single-layer uterine closure after a first CS. The study protocol and the effect paper have been published elsewhere[17 18] No substantial changes were made to the protocol after commencement of the trial.[17] This trial-based economic evaluation is reported according to the Consolidated Health Economic Evaluating Reporting Standards statement.[19]

### Target population
All women who underwent a first CS, planned or unplanned, at one of the participating hospitals were asked to participate in the study. Inclusion criteria were sufficient command of the Dutch or English language, age 18 years or older and written informed consent. Exclusion criteria were: inadequate possibility for counselling (eg, indication for emergency CS without being informed about the study previously, women in severe pain without adequate therapy), previous major uterine surgery (eg, laparoscopic or laparotomic fibroid resection, septum resection), women with known causes of menstrual disorders (eg, cervical dysplasia, communicating hydrosalpinx, uterine anomaly or endocrine disorders disturbing ovulation), placenta increta or percreta during the current pregnancy, or three or more fetuses during the current pregnancy. After informed consent had been signed and a CS was indicated, participants were randomly allocated to receive single-layer (control) or double-layer (intervention) closure of the uterine incision in a 1:1 ratio. Due to the nature of the treatment, surgeons performing the CS were not masked to the allocated method. Participants and sonographers were blinded to the allocation, researchers and statisticians were not. Detailed information about study design and randomisation can be found in the study protocol.[17]

### Choice of health outcomes
For this trial-based economic evaluation, two main health outcomes were used: postmenstrual spotting (referred to as spotting days in this paper) and quality-adjusted life-years (QALYs) at nine months after CS. Spotting days was chosen because it has been strongly related to a niche[5 6] (ie, an indentation at the site of the caesarean scar with a depth of at least 2 mm[20]), which may be influenced by uterine closure technique.[8] QALY is routinely used as a summary outcome measure of health in economic evaluations, because it incorporates the impact of interventions on both the quantity and quality of life,[21] and allows decision-makers to compare the effectiveness and CE of a range of interventions for different health conditions.[22]

### Study perspective and time horizon
This trial-based economic evaluation was performed from a societal and a healthcare perspective over a time horizon of nine months. Therefore, discounting of costs and effects was not necessary. When a healthcare perspective is adopted, only the intervention costs and costs related to healthcare utilisation are included in the analysis.[23] For the societal perspective, costs related to informal care and productivity losses are included in addition to intervention and healthcare utilisation costs.[23]

### Setting and location
In total, 32 hospitals in the Netherlands, both academic (n=6) and non-academic (n=26), collaborating within the Dutch Consortium for Healthcare Evaluation and Research in Obstetrics and Gynaecology (Consortium 2.0, www.zorgevaluatienederland.nl), participated in this study.[17] In the Netherlands, a CS is only performed in a hospital setting. In most cases, without maternal or

neonatal complications, women will be discharged from the hospital after a CS after two or three days. All costs regarding the CS and admission days are standard care and are paid by an individual's health insurance. Maternity leave of at least ten weeks is regulated through the Employee Insurance Agency. Paternity leave is limited to one week. The first eight days after delivery, a maternity nurse visits the family at least three hours a day.

### Control and intervention condition

The control group underwent single-layer closure of the uterus using unlocked continuous running multifilament sutures, which is the usual care provided by hospitals in the Netherlands.[17] In the intervention group, double-layer closure of the uterine incision was performed using unlocked multifilament continuous running sutures for both layers and the endometrial layer was included in the first layer. The second layer was a continuous running suture that imbricated the first. A mandatory online instruction video was shown to all surgeons in participating hospitals prior to participation for the intervention group. The exact procedures in both study arms regarding uterine closure are described in the study protocol.[17]

At baseline, data were collected on sociodemographic characteristics for all participants.[17]

### Outcomes

#### Health outcomes

Spotting days was the primary outcome of the trial, and was defined as number of days with brownish discharge for more than two days at the end of the menstruation, with a total duration (menstruation and spotting) of more than seven days, or intermenstrual blood loss that started after the end of the menstruation.[5] Spotting days were self-reported by participants through a digital questionnaire at nine months after CS, including a calendar on which women could record daily blood loss during one month. Women who reported that they had no blood loss were classified as amenorrhoeic.

Health-related quality of life was measured using the EuroQol five dimensions five levels (EQ-5D-5L) at baseline, and at three and nine months after CS.[24] The EQ-5D-5L has five dimensions of quality of life (mobility, self-care, usual activities, pain/discomfort and anxiety/depression) using five response levels (ie, no problems, slight problems, moderate problems, severe problems or extreme problems) describing 3125 health states.[24] The participants' health states obtained from EQ-5D-5L responses were converted into utility values using the Dutch tariff.[25] The utility values were used to calculate QALYs by means of the area under the curve method (ie, the duration of a health state is multiplied by the utility related to that health state).

### Cost outcomes

#### Intervention costs

The average costs of performing a CS reported by the participating hospitals was €5360. The intervention incurred additional suture material and additional operation time (3.9 minutes on average). The costs of additional resources were obtained from the academic and non-academic hospitals using a bottom-up approach. On average, the intervention resulted in additional costs of €95.79 per participant in academic hospitals and €71.14 per participant in non-academic hospitals (online supplemental table S1).

#### Healthcare utilisation and informal care costs

A specifically adapted version of the iMTA Medical Cost Questionnaire (iMCQ)[26] was used to measure healthcare utilisation and care provided by family and/or friends (ie, informal care) using 3-month and 6-month recall periods at three and nine months of follow-up, respectively. The iMCQ is a standardised generic instrument for measuring medical costs including questions related to healthcare utilisation and informal care.[26] Healthcare utilisation was valued using prices from the Dutch costing guideline.[27] Healthcare utilisation costs included primary care costs (eg, costs of visits to general practitioners, health professionals and complementary healthcare providers), secondary care costs (eg, costs of ambulatory hospital visits, visits to other healthcare organisations and admissions to the hospital), and medication costs. Secondary care costs were recorded after discharge from the hospital. The average costs of performing the CS included the operation and hospital stay until discharge and were, therefore, not included in the secondary care costs to avoid double counting.

The informal care costs were based on the amount of time the participant needed help in performing household tasks or received care from family and/or friends, because of health problems. Dutch standard prices were used for informal care costs.[27] Medication use was valued using data from the Dutch Healthcare Institute (www.medicijnkosten.nl).[28]

#### Lost productivity costs

The iMTA Productivity Cost Questionnaire (iPCQ)[29] was used to measure self-reported sickness absenteeism from paid and unpaid work, and presenteeism using 3-month and 6-month recall periods at three and nine months of follow-up, respectively. The iPCQ is a standardised generic questionnaire to measure productivity costs and it is applicable to national and international studies.[29] The friction cost approach (FCA) was used to calculate sickness absenteeism costs from paid work.[30] The FCA assumes that sickness absenteeism costs are limited to the period needed to replace an absent, sick worker (the friction period), which has been estimated to be 12 weeks (85 days) in the Netherlands.[27] Gender-specific estimates of the mean wages of the Dutch population were used to calculate sickness absenteeism costs from paid work.[27]

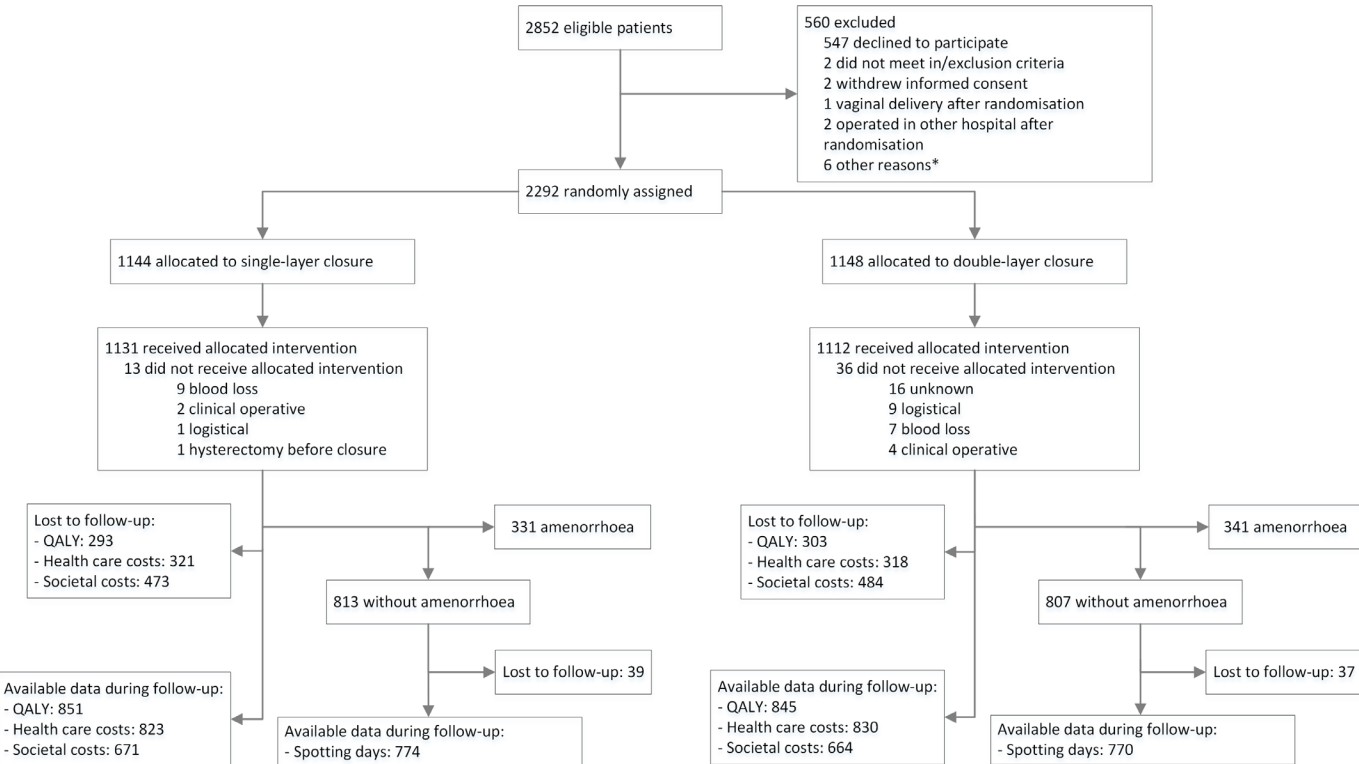

**Figure 1** Trial profile. *Logistical reasons, computer randomisation issues, passing through the allocated method to operating gynaecologist, or participant not traceable after randomisation. QALYs, quality-adjusted life-years.

To measure sickness absenteeism from unpaid work, the participants were asked whether they had difficulty in performing unpaid work activities due to sickness (eg, household tasks, childcare, voluntary work), and if that was the case, for how many hours.[29] Costs related to sickness absenteeism from unpaid work were valued using a shadow price for legally employing a domestic assistant.[29] To measure participants' level of presenteeism, participants rated how efficiently they worked while suffering from health complaints on a scale from 0 (I was unable to do anything) to 10 (I was able to do as much as usual). The resulting efficiency score was used to calculate presenteeism costs: Presenteeism costs=number of days working with complaints * [1 - (efficiency score / 10)] * number of working hours per day * gender-specific mean wage rates.[31]

### Statistical analysis

The main analyses included all participants with a menstrual cycle (ie, participants with amenorrhoea for any reason were excluded) at nine months follow-up.

Analyses were performed according to the intention-to-treat principle using StataSE V.16 (Stata). Multiple imputation by chained equations, stratified by group allocation, was used to impute missing data. Variables associated with missingness and outcomes as well as potential confounders were included in the imputation model such as age, body mass index, level of education, parity, previous miscarriage, gestational age at CS, hypertension, diabetes, use of contraception and breast feeding.

Predictive mean matching was used in the imputation procedure to account for the skewed distribution of the costs.[32] The number of imputations was increased until there was a loss of efficiency of ≤5%, resulting in 20 imputed datasets.[32 33] The 20 datasets were analysed separately and estimates were pooled using Rubin's rules.[34] After multiple imputation, amenorrhoeic women were excluded from the analyses as a priori decided, because the outcome spotting days could not be evaluated in these women.[17]

Differences in costs and effects between treatment groups at 9 months follow-up were estimated using seemingly unrelated regression analyses, which accounts for the correlation between costs and effects.[35] The intraclass correlation coefficient (ICC, ie, the variation around the subjects belonging to the same hospital cluster divided by the total variance between hospitals)[36] was small (ICC=0.004). This means that hardly any of the variance in the outcome measure was accounted for by clustering at the level of the hospital. In addition to the small ICC, patients were randomised at the individual level and not at the hospital level. Therefore, multilevel analysis was not necessary. Incremental cost-effectiveness ratios were calculated by dividing the difference in costs (ie, total societal costs and total healthcare costs) between groups by the difference in effects.

Bias-corrected accelerated bootstrapping with 5000 replications was used to estimate the joint uncertainty surrounding differences in costs and effects.

Bootstrapped cost-effect pairs were plotted on CE planes.[37] Cost-effectiveness acceptability curves were estimated that show the probability of double-layer closure being cost-effective compared with single-layer closure for a range of willingness-to-pay (WTP) thresholds (ie, the maximum amount of money society is willing to pay for a unit of effect gained).[38] For spotting days, we used a maximum WTP threshold of €253 per one day reduction. This threshold was based on the value of 8 hours of paid work given the average productivity costs per working hour for women in the Netherlands (ie, €31.6 per hour). For QALYs, we used a WTP threshold of €20 000/QALY gained recommended by the Dutch Healthcare Institute[27] and €23 420/QALY (equivalent of £20 000/QALY) recommended by the National Institute for Health and Clinical Excellence.[39]

## Sensitivity analysis

Four sensitivity analyses (SA) were performed to assess the robustness of the results. SA1 consisted of a cost-effectiveness analysis (CEA) including all women randomised (ie, without excluding amenorrhoeic women) from both a societal and healthcare perspective for the QALY outcome. SA2 consisted of a complete case analysis from both a societal and healthcare perspective including only women without amenorrhoea for both spotting days and QALYs. Third, we performed per protocol analyses for both outcomes from both a societal and healthcare perspective (SA3). Finally, we performed a SA in which we adjusted for hormonal contraception and breast feeding

(exclusively or combined with formula) at nine months of follow-up (SA4).

## Patient and public involvement

The Dutch gynaecological patients' association agreed on the design of the study and the grant proposal for funding. They were not involved in outcome measures or recruitment, and they were not asked to give advice in the interpretation of the results. We will disseminate the study results to all participants, and to the public through popular science articles.

## RESULTS

### Participants

In total, 2292 women undergoing a first CS were included. Of them, 1144 participants were randomised to single-layer and 1148 participants to double-layer closure of the uterine incision. In the single-layer group, 694 (60.7%) participants underwent planned CS, and in the double-layer group this was done in 705 (61.4%) participants.

In total, 672 women reported amenorrhoea (n=331 in control and n=341 in intervention group), resulting in 1620 women reported having menstrual blood loss over 9 months of follow-up and were included in the main analyses (n=813 in control and n=807 in intervention group) (figure 1). Of all women included in the main analysis (n=1620), 95% had completed follow-up data for spotting days (n=1544, 774 in control group and 770 in intervention group). Within the total group (n=2292), complete follow-up data were available for 74% of QALYs (n=1696, 851 in control group and 845 in intervention group), for 72% of total healthcare costs (n=1653, 823 in control group and 830 in intervention group), and for 58% of total societal costs (n=1335, 671 in control group and 664 in intervention group) (figure 1). At baseline, no meaningful differences were found between both groups (table 1). At nine months follow-up, 12.1% of women in the single-layer arm and 17.1% of women in the double-layer arm were breastfeeding their children. In the single-layer arm, 40.3% of the participants used hormonal contraceptives at nine months follow-up, and in the double-layer arm this was 38.2%. Participants with complete follow-up were more likely to be nulliparous and to have a higher education level compared with participants without complete follow-up.

### Effectiveness

There were no statistically significant differences between groups in spotting days (mean difference −0.056, 95% CI −0.374 to 0.263) and QALYs (mean difference −0.005, 95% CI −0.015 to 0.005) at nine months follow-up (table 2).

### Costs

The main contributors to total societal costs in both groups were lost productivity costs (€5689 in control group and €5927 in intervention group) and healthcare

**Table 1** Baseline characteristics of women without amenorrhoea in the control group and intervention group

|  | Single layer (n=813)* | Double layer (n=807)† |
|---|---|---|
| Age, years | 32.1 (4.7) | 32.0 (4.6) |
| Level of education | | |
| Low | 50 (6.5) | 54 (7.1) |
| Middle | 263 (34.2) | 242 (31.8) |
| High | 452 (58.8) | 457 (60.0) |
| Nulliparous women | 568 (73.9) | 578 (75.9) |
| BMI (kg/m$^2$) | 26.4 (4.5) | 26.7 (4.9) |
| Smoking habit | 44 (5.7) | 37 (4.9) |
| Hypertension | 146 (19.0) | 127 (16.7) |
| Diabetes mellitus | 89 (11.6) | 66 (8.7) |
| Gestational age | 38.6 (2.4) | 38.6 (2.3) |
| Previous miscarriage | 255 (33.2) | 221 (29.0) |
| Previous ectopic pregnancy | 10 (1.3) | 12 (1.6) |
| Planned CS | 504 (62.0) | 503 (62.3) |

Data are mean (SD) or n (%). N is equal to the total number of patients in the group.
*5.5% missing data for all variables, except 'planned CS' (0%).
†5.9% missing data for all variables, except 'planned CS' (0%).
BMI, body mass index; CS, caesarean section.

**Table 2** Multiply imputed mean effects and costs by group and mean difference at 9 months follow-up in women without amenorrhoea

| | Single layer (n=813) | Double layer (n=807) | Mean difference* (95% CI) |
|---|---|---|---|
| **Effects** | | | |
| Spotting days | 1.44 (0.11) | 1.39 (0.11) | −0.056 (−0.374 to 0.263) |
| QALYs gained | 0.663 (0.003) | 0.658 (0.004) | −0.005 (−0.015 to 0.005) |
| **Costs** | | | |
| Intervention costs† | 0 | 76 (0.31) | 76 (75 to 76) |
| Primary care costs | 255 (16) | 250 (17) | −5 (−49 to 40) |
| Secondary care costs | 400 (75) | 317 (44) | −83 (−292 to 38) |
| Medication costs | 89 (84) | 84 (23) | −5 (−103 to 70) |
| Total healthcare costs‡ | 744 (112) | 727 (58) | −17 (−273 to 143) |
| Informal care costs | 77 (18) | 124 (33) | 47 (−10 to 141) |
| Absenteeism costs at paid work | 1052 (122) | 1009 (110) | −42 (−34 to 261) |
| Absenteeism costs at unpaid work | 3525 (226) | 3810 (263) | 284 (−360 to 964) |
| Presenteeism costs | 290 (26) | 256 (24) | −34 (−98 to 28) |
| Total lost productivity costs | 4857 (280) | 5076 (299) | 208 (−574 to 999) |
| **Total societal costs§** | 5689 (321) | 5927 (324) | 238 (−624 to 1108) |

Data are mean (SE). Multiple imputation model consisted of age, education level, parity, body mass index, smoking habit, hypertensive disorder, diabetic status, gestational complications, gestational age, previous miscarriage or ectopic pregnancies, use of contraception, breastfeeding and self-reported menstrual blood loss.
Primary care: costs of visits to general practitioners, health professionals, and complementary healthcare providers. Secondary care: costs of ambulatory hospital visits, visits to other healthcare organisations and hospital admissions. Medication costs: costs of medication use after discharge from the hospital. Informal care costs: costs of received care from family and/or friends due to health problems. Absenteeism costs at paid work: costs of sickness absenteeism from paid work. Absenteeism costs at unpaid work: costs of absenteeism from unpaid work activities (eg, household tasks, childcare, voluntary work). Presenteeism costs: costs of working while suffering from health complaints.
*Cost and effect differences at nine months follow-up were estimated using seemingly unrelated regression analyses.[35]
†Additional intervention costs to perform double-layer, excluding the average costs for performing a caesarean section (€5360,-).
‡The sum of intervention, primary care, secondary care and medication costs.
§The sum of total healthcare costs, informal care costs and lost productivity costs.
QALY, quality-adjusted life-years.

costs (€744 in control group and €727 in intervention group, table 2). There were no differences in days of hospital stay between the groups.[18] Costs associated with absenteeism from unpaid work due to sickness (ie, costs related to getting help to perform household tasks, childcare, voluntary work) was the largest contributor to lost productivity costs in both groups (€3525 in control group and €3810 in intervention group). Secondary care costs were the largest contributor to total healthcare costs in both groups (€400 in control, group and €317 in intervention group). During the follow-up, there were no statistically significant differences between groups in total healthcare costs (€−17, 95% CI −273 to 143) and total societal costs (€238, 95% CI −624 to 1108, table 2). Presenteeism costs were not statistically significantly different in the intervention group compared with the control group (€−34, 95% CI −98 to 28, table 2).

## CE analysis

From a societal perspective, most bootstrapped cost-effect pairs (44%) were in the North East Quadrant of the CE-plane for spotting days (table 3, figure 2A). The probability of double-layer closure being cost-effective compared with single-layer was 0.30 at a WTP of €0/ spotting day less and 0.31 at €253/spotting day less (figure 2B). For QALYs, most of the bootstrapped cost-effect pairs (62%) was in the North West Quadrant of the CE-plane (table 3, figure 2C). The probability of double-layer closure being cost-effective compared with single-layer at both the Dutch WTP threshold of €20 000/QALY gained, and the UK WTP threshold of €23 420/QALY gained, was 0.25 from a societal perspective (figure 2D, online supplemental table S2).

From a healthcare perspective, bootstrapped cost-effect pairs were equally distributed among the Eastern and Western quadrants of the CE-plane for spotting days (table 3, figure 2E). This shows that uncertainty around costs and effects is large. The CEAC presented in figure 2E2F shows that if the WTP for one spotting day less is €0, the probability of double-layer closure being cost-effective in comparison with single-layer was 0.55. This probability increases to 0.59 if the WTP is €253/spotting day less (online supplemental table S2). For QALYs, from

**Table 3** Results of the cost-effectiveness analysis

| Effect outcome* | Cost difference, € (95% CI) | Effect difference (95% CI) | ICER €/ effect gained | Distribution of the cost-effectiveness plane | | | |
|---|---|---|---|---|---|---|---|
| | | | | North East | South East | South West | North West |
| **Main analysis—societal perspective** | | | | | | | |
| Spotting days | 238 (−624 to 1108) | 0.056 (−0.263 to 0.374) | 4281 | 44% | 20% | 10% | 26% |
| QALY | 238 (−624 to 1108) | −0.005 (−0.015 to 0.005) | −49699 | 8% | 9% | 21% | 62% |
| **Main analysis—healthcare perspective** | | | | | | | |
| Spotting days | −17 (−283 to 146) | 0.056 (−0.263 to 0.374) | −311 | 30% | 34% | 18% | 18% |
| QALY | −17 (−283 to 146) | −0.005 (−0.015 to 0.005) | 3614 | 6% | 10% | 42% | 42% |
| **SA1—including all women randomised—societal perspective** | | | | | | | |
| QALY | 150 (−764 to 944) | −0.006 (−0.014 to 0.002) | −25696 | 4% | 6% | 30% | 60% |
| **SA1—including all women randomised—healthcare perspective** | | | | | | | |
| QALY | −235 (−1230 to 84) | −0.005 (−0.014 to 0.004) | 46765 | 3% | 10% | 66% | 21% |
| **SA2—complete-case analysis—societal perspective** | | | | | | | |
| Spotting days | 346 (−641 to 1394) | 0.149 (−0.452 to 0.138) | 2324 | 69% | 18% | 3% | 10% |
| QALY | 313 (−671 to 1382) | 0.006 (−0.016 to 0.003) | 50787 | 73% | 17% | 5% | 5% |
| **SA2—complete-case analysis—healthcare perspective** | | | | | | | |
| Spotting days | −9 (−251 to 174) | 0.186 (−0.486 to 0.103) | −46 | 54% | 37% | 4% | 5% |
| QALY | 1 (−256 to 172) | 0.007 (−0.016 to 0.002) | 80 | 59% | 34% | 4% | 3% |
| **SA3—per-protocol analysis—societal perspective** | | | | | | | |
| Spotting days | 43 (−820 to 903) | 0.043 (−0.274 to 0.361) | 1008 | 32% | 29% | 17% | 22% |
| QALY | 43 (−820 to 903) | 0.004 (−0.006 to 0.013) | 11909 | 47% | 31% | 15% | 7% |
| **SA3—per-protocol analysis—healthcare perspective** | | | | | | | |
| Spotting days | −39 (−301 to 126) | 0.043 (−0.2748 to 0.361) | −909 | 23% | 38% | 23% | 16% |
| QALY | −39 (−301 to 126) | 0.004 (−0.006 to 0.013) | −10745 | 32% | 45% | 16% | 7% |
| **SA4—main analysis adjusted—societal perspective** | | | | | | | |
| Spotting days | 226 (−633 to 1092) | 0.046 (−0.277 to 0.369) | 4961 | 41% | 19% | 12% | 28% |
| QALY | 226 (−633 to 1092) | −0.004 (−0.014 to 0.006) | −58497 | 10% | 11% | 20% | 59% |
| **SA4—main analysis adjusted—healthcare perspective** | | | | | | | |
| Spotting days | −11 (−267 to 154) | 0.046 (−0.277 to 0.369) | −248 | 29% | 31% | 19% | 21% |
| QALY | −11 (−267 to 154) | −0.004 (−0.014 to 0.006) | 2925 | 9% | 13% | 37% | 41% |

Data are mean (95% CI).
Main analysis: CEA from a societal and a healthcare perspective for spotting days and QALY including only women without amenorrhoea (total=1620, control n=813, intervention n=807).
SA1: CEA from a societal and a healthcare perspective for QALY, including all women randomised in the study after multiple imputation (ie, without excluding amenorrhoeic women, n=2292) (online supplemental table S3)).
SA2: CEA from a societal perspective using complete cases for spotting days and total societal costs (total=1065, control n=544, intervention n=521) including only women without amenorrhoea (n=1620).
SA2: CEA from a societal perspective using complete cases for QALY and total societal costs (total=1057, control n=541, intervention n=516) including only women without amenorrhoea (n=1620).
SA2: CEA from a healthcare perspective using complete cases for spotting days and total healthcare costs (total=1315, control n=662, intervention n=653) including only women without amenorrhoea (n=1620).
SA2: CEA from a healthcare perspective using complete cases for QALY and total healthcare costs (total=1310, control n=657, intervention n=653) including only women without amenorrhoea (n=1620).
SA3: per-protocol analysis for spotting days and QALY from a societal perspective (total=1620, control n=828, intervention n=792) including only women without amenorrhoea (n=1620).
SA3: per-protocol analysis for spotting days and QALY from a healthcare perspective (total=1620, control n=828, intervention n=792) including only women without amenorrhoea (n=1620).
SA4: main analysis adjusted for the use of contraception and breastfeeding during follow-up from a societal and a healthcare perspective (total=1620, control n=813, intervention n=807).
*The effect outcome 'spotting days' was multiplied by −1 in the cost-effectiveness analysis to keep the CE-plane interpretable.
CE, cost-effectiveness; CEA, cost-effectiveness analysis; ICER, incremental cost-effectiveness ratio; QALY, quality-adjusted life-year; SA, sensitivity analysis.

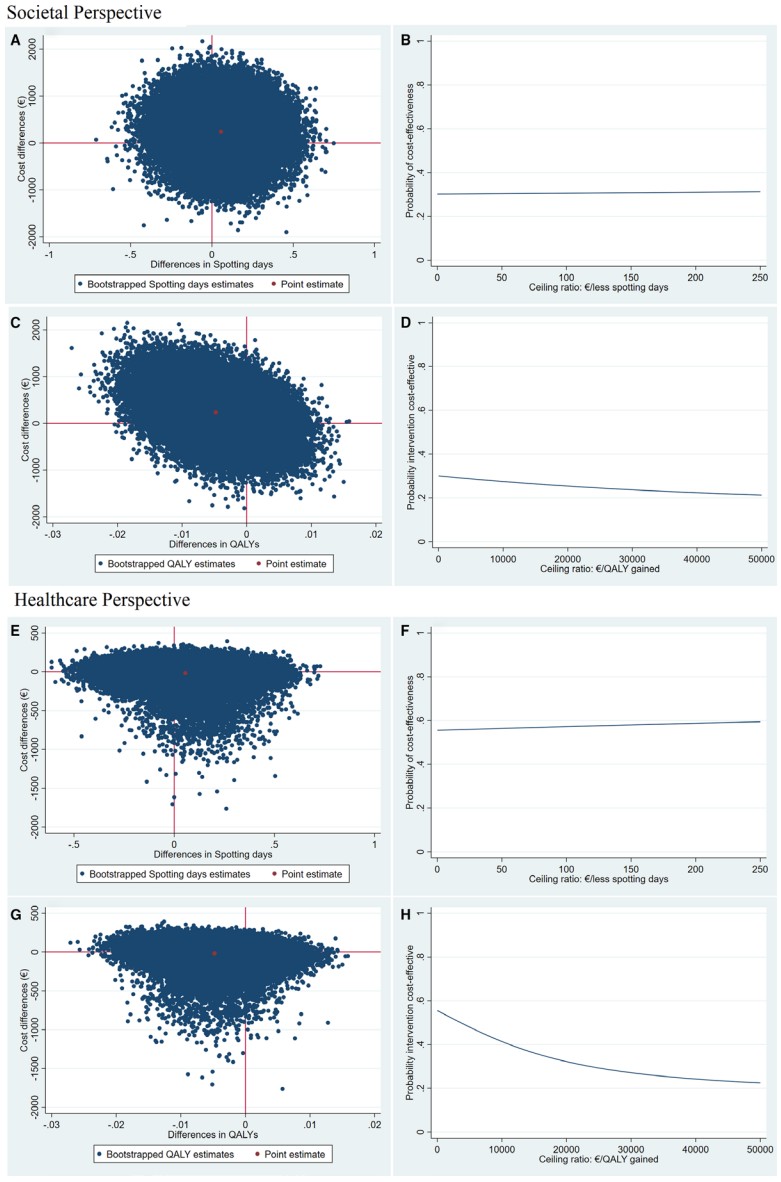

**Figure 2** Cost-effectiveness planes and cost acceptability curves from a societal and healthcare perspective comparing double-layer to single-layer uterine closure. (1) Cost-effectiveness plane (CE plane) showing the incremental cost-effectiveness ratio point estimate (ICER, red dot) and the distribution of the 5000 replications of the bootstrapped cost-effective pairs (blue dots). (2) Cost-effectiveness acceptability curve (CEAC) indicating the probability of double-layer uterine closure being cost-effective compared with single-layer closure (y-axis) for different willingness-to-pay (WTP) thresholds per unit of effect gained (x-axis). (A) CE plane for spotting days from a societal perspective showing that most of bootstrapped cost-effect pairs were equally distributed across CE plane quadrants representing high uncertainty around ICER. (B) CEAC for spotting days from a societal perspective indicating a steady 0.2 probability of double-layer uterine closure being cost-effective compared with single-layer closure for different WTP thresholds per fewer spotting days. (C) CE plane for QALYs from a societal perspective showing that most of the bootstrapped cost-effect pairs were in the Northern quadrants (ie, higher costs) and Western quadrants where double-layer uterine closure was less effective compared with single-layer closure. (D) CEAC for QALYs from a societal perspective indicating a probability of double-layer uterine closure being cost-effective around 0.2 for different WTP thresholds per QALY gained. (E) CE plane for spotting days from a healthcare perspective showing that most of the bootstrapped cost-effect pairs were in Southern quadrants, where double-layer uterine closure was less costly compared with single-layer closure, but they are equally distributed across the Eastern and Western quadrants representing high uncertainty around the effectiveness of double-layer uterine closure compared with single-layer closure. (F) CEAC for spotting days from a healthcare perspective indicating a steady 0.6 probability of double-layer uterine closure being cost-effective compared with single-layer closure for different WTP thresholds per fewer spotting days. (G) CE-plane for QALYs from a healthcare perspective showing that most of the bootstrapped cost-effect pairs were in the Southern quadrants (ie, lower costs) and Western quadrants where double-layer uterine closure was less effective compared with single-layer closure. (H) CEAC for QALYs from a healthcare perspective indicating that the probability of double-layer uterine closure being cost-effective compared with single-layer closure decreased with an increasing of the different WTP thresholds per QALY gained because healthcare costs were on average lower in the intervention group while it is less effective compared with the usual practice.

a healthcare perspective, most of bootstrapped cost-effect pairs were in the North West Quadrants of the CE-plane (table 3, figure 2G). The probability of double-layer closure being cost-effective compared with single-layer closure at the Dutch and UK WTP threshold (€20 000 and €23 420/QALY gained, respectively) was 0.41 from a healthcare perspective (table 3, figure 2G2H).

The results of the SA were similar to those of the main analysis (table 3).

## DISCUSSION

The results of this trial-based economic evaluation showed that double-layer uterine closure after a first CS did not significantly decrease spotting days nor improve QALYs compared with single-layer closure at nine months follow-up. In addition, total healthcare costs and societal costs related to double-layer closure did not significantly differ from single-layer closure. Low probabilities of double-layer closure being cost-effective in comparison with single-layer closure were found for all relevant WTP thresholds. Therefore, double-layer closure was not considered cost-effective compared with single-layer closure after a first CS from a societal and a healthcare perspective.

### Comparison with previous studies

The results of this economic evaluation are not in line with our hypothesis, which was based on previously conducted observational studies[5 6 40] and meta-analyses.[14] These showed that single-layer closure resulted in thinner residual myometrium and a higher proportion of large niches than double-layer closure. These sonographical findings, or surrogates, were suggested to lead to more postmenstrual spotting and, therefore, higher related costs. Although double-layer closure resulted in increased CS costs, these costs were neutralised by higher secondary care and presenteeism costs in the single-layer closure group, resulting in no overall difference in total healthcare costs or total societal costs.

To the best of our knowledge, this is the first study evaluating the CE of double-layer closure in comparison with single-layer closure after a CS. The largest study (n=15 935) on this topic mentioned possible cost savings but no CE analysis was performed.[41] The second largest study (n=3033) comparing single-layer versus double-layer closure in a factorial randomised controlled trial hypothesised on a possible reduction in costs in their study protocol, since a CS is conducted so frequently that 'any difference in morbidity is likely to have significant cost and community effects'. The authors found a difference in operative time, though they did not discuss costs in the 2010 publication, and it is unlikely that a CE analysis is going to be performed.[42]

Based on the current and previous studies,[43 44] we recommend to leave the choice of uterine closure technique with the preference of the surgeon. Previous studies reported only short-term maternal outcomes in the first few weeks after CS.[42–44] We confirmed these findings using a follow-up of nine months after CS. In addition, we showed that there is no difference in costs between the two types of closure. Our three years follow-up results will show whether double-layer is superior compared with single-layer closure with regard to long-term outcomes. These outcomes include fertility outcomes, pregnancy complications and mode of delivery, as well as safety outcomes such as uterine dehiscence or rupture and related neonatal and maternal morbidity. In addition, a CE analysis for long-term outcomes will be performed as well. When superiority cannot be shown on the long-term either, guidelines should recommend to leave the uterine closure technique regarding single-layer versus double-layer up to the preference of the performing surgeon.

### Strengths and limitations

This study was performed alongside a large multicentre randomised controlled superiority trial, which is considered the best vehicle for economic evaluations because it allows the prospective collection of cost and effect data and the use of patient level information for drawing inferences about additional costs and benefits of interventions.[16] Additionally, the CEA was conducted from a societal perspective meaning that all relevant costs for decision making (ie, intervention, healthcare utilisation, informal care and lost productivity costs) were included in the analysis.[22] Several SAs were performed, to assess the robustness of our results, which resulted in similar results as compared with the main analyses.

However, one of the limitations of this study was that the cost questionnaires included retrospective self-reported questions over a 3-month and 6-month period, which may have caused recall bias. Nevertheless, we assume that this bias is equally distributed across the two groups and does, therefore, does not impact the difference between groups. Although there is no gold standard for measuring lost productivity costs, we used a standardised instrument,[29] which is considered best practice currently.[45] Another limitation is that generalisability of the results to healthcare systems in other countries may be limited, as they may adopt different usual practices and have different payment systems.[46] Additionally, generalisability may be impaired since in our study sample relatively many planned CS were performed compared with the Dutch average,[47] probably resulting in an overall underestimation of niche related postmenstrual spotting.

### Future research

It is important to realise that we have only evaluated the aspect of single-layer versus double-layer closure of the uterine incision in our trial. A CS consists of multiple steps and other aspects of the surgical technique used to perform a CS may also affect clinical outcomes and costs, and should therefore be subject to future research. Examples of uterine incision and repair are the level of hysterotomy (above or below the plica vesicouterina)[48]

and inclusion or exclusion of the endometrium in the uterine suture.

## Conclusions and policy implications

In conclusion, double-layer uterine closure is not cost-effective compared with single-layer uterine closure from both a societal and healthcare perspective. Thus, from a CE point of view, there is no reason to advocate double-layer over single-layer uterine closure. Long-term follow-up will show whether guidelines should be adapted based on obstetric and reproductive outcomes of double-layer closure compared with single-layer closure.

**Author affiliations**
[1]Obstetrics and Gynaecology, Amsterdam Reproduction & Development, Amsterdam UMC, Vrije Universiteit Amsterdam, Amsterdam, the Netherlands
[2]Health Sciences, Amsterdam Public Health, Faculty of Science, Vrije Universiteit Amsterdam, Amsterdam, the Netherlands
[3]Obstetrics and Gynaecology, Deventer Ziekenhuis, Deventer, the Netherlands

**Acknowledgements** We would like to thank all participants of the 2Close study. Additionally, we would like to thank the Departments of Obstetrics and Gynaecology of all participating hospitals, with special thanks to all research nurses and research midwives for their contribution in data collection.

**Collaborators** 2Close study group: WM (Marchien) van Baal, Erik van Beek, Mireille N Bekker, Ângela Jornada Ben, Karin de Boer, Elisabeth MA Boormans, Judith E Bosmans, Hugo WF van Eijndhoven, Mohamed El Alili, AH (Hanneke) Feitsma, Christianne JM de Groot, Majoie Hemelaar, Wouter JK Hehenkamp, Wietske Hermes, Esther Hink, Judith AF Huirne, Anjoke JM Huisjes, CAH (Ineke) Janssen, Kitty Kapiteijn, Mesrure Kaplan, Paul JM van Kesteren, Judith OEH van Laar, Josje Langenveld, Wouter J Meijer, Angèle LM Oei, Eva Pajkrt, Dimitri NM Papatsonis, Celine M Radder, Robbert JP Rijnders, HCJ (Liesbeth) Scheepers, Daniela H Schippers, Nico WE Schuitemaker, Sanne I Stegwee, Marieke Sueters, Harry Visser, Huib AAM van Vliet, LHM (Marloes) de Vleeschouwer, Lucet F van der Voet.

**Contributors** SS collected data, interpreted data and drafted the report. AJB and MEA analysed and interpreted data and drafted the report. LvdV designed the study, collected data and revised the first draft of the report critically. CdG designed the study and revised the first draft of the report critically. JB designed the study, interpreted data and participated in drafting and revising the report. JH designed the study, was principle investigator, interpreted the data and participated in drafting and revising the report. All other members of the 2Close study group (WMvB, EvB, MNB, KdB, EMAB, HWFvE, AHF, MH, WJKH, WH, EH, AJMH, CAHJ, KK, MK, PJMvK, JOEHvL, JL, WJM, ALMO, EP, DNMP, CMR, RJPR, HCJS, DHS, NWES, MS, HV, HAAMvV, LHMdV) agreed with the design of the trial, participated in data collection as local investigators, and revised the draft paper. All authors and collaborators approved the final version of the manuscript.

**Funding** This study was performed with funding from ZonMw: The Netherlands Organisation for Health Research and Development (project number 843002605). The funder of the study approved the study protocol.

**Disclaimer** The funder had no role in study design, data collection, data analysis, data interpretation, or writing of the manuscript. The corresponding author had full access to all the data in the study and had final responsibility for the decision to submit for publication.

**Competing interests** JH received grants from ZonMw, during the conduct of the study; and reports grants from Samsung, grants from PlantTec Medical, and received a fee from Olympus, all outside the submitted work. CdG received a grant from ZonMw outside the submitted work.

**Patient consent for publication** Not required.

**Ethics approval** The study was approved by the Institutional Review Board (IRB) of Amsterdam UMC-location VU University medical centre in December 2015 (registration number 2015.462) and by the boards of all participating hospitals before start of inclusion. No substantial changes were made to the protocol after commencement of the trial. All participants provided written informed consent before taking part in the study.

**Provenance and peer review** Not commissioned; externally peer reviewed.

**Data availability statement** Data are available upon reasonable request. Data sharing: De-identified individual participant data collected during the 2Close trial will be shared at one year after publication of the long-term results on request (j.huirne@amsterdamumc.nl). Approval of a proposal will be necessary before data will be shared. To gain access, requesters will need to sign an agreement form and confirm that data will be used for the purpose for which access was granted.

**ORCID iDs**
Sanne I. Stegwee http://orcid.org/0000-0003-1353-5576
Ângela J. Ben http://orcid.org/0000-0003-4793-9026
Mohamed El Alili http://orcid.org/0000-0002-6341-7976
Lucet F. van der Voet http://orcid.org/0000-0001-5389-646X
Christianne J.M. de Groot http://orcid.org/0000-0003-3277-2542
Judith E. Bosmans http://orcid.org/0000-0002-1443-1026
Judith A.F. Huirne http://orcid.org/0000-0002-8248-2677

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
