## [Reviewer comments · BMJ Open]

ARTICLE DETAILS

TITLE (PROVISIONAL)	Cost-effectiveness of single- versus double-layer uterine closure during caesarean section on postmenstrual spotting: economic evaluation alongside a randomised controlled trial
AUTHORS	Stegwee, Sanne; Ben, Ángela; El Alili, Mohamed; van der Voet, Lucet; de Groot, Christianne; Bosmans, J; Huirne, Judith

VERSION 1 – REVIEW

REVIEWER	Anna Joy Rogers University of Tennessee Health Science Center, Tennessee, USA
REVIEW RETURNED	28-Oct-2020

GENERAL COMMENTS	This is an economic evaluation alongside a superiority trial comparing double-layer to single-layer uterine closure during a primary CS. The study is well designed and executed. The CEA is performed to highest current standards. The justification for choice of outcome, as well as discussion of mechanisms by which the intervention is proposed to improve outcomes, needs to be strengthened. This could impact how the results are presented. ***INTRODUCTION*** 62: consider also morbidity from cesarean scar pregnancies 65,66: both statements need citations 67: over what timeline do 30% of women with a niche develop AUB? Please make a stronger argument for why spotting days in the first nine months is an important outcome. Additionally, is reducing spotting days the only mechanism by which you hypothesize a double-layer closure is superior to a single-layer closure in the first nine months? If so, that changes a great deal about how your analysis should be conducted, because you should leave out all the amenorrheic women from your analyses. Leaving them in would only add random variation to outcome and cost estimates. Please consider grammatical review. ***METHODS*** In the introduction, you state that sonographically visible indentation is visible at site of previous uterine incision, and that this niche is associated with AUB. You never make the connection that double versus single uterine closure is associated with lower risk of a niche. That seems to be an important mediator of uterine bleeding. Is this investigated in the trial? Study design – appropriate. According to current economic evaluation standards. Target population – inclusion and exclusion criteria appears
--

	appropriate Health outcomes – To my understanding, spotting or irregular menstruation during the postpartum return of menses is a common phenomenon regardless of mode of delivery. Women may expect some measure of menstrual irregularity and may not see a physician for this. Additionally, their mode of child feeding (exclusive breast, formula, or mixture of both) and postpartum contraceptive method (IUD, Implant, OCPs, Depo) all could have a large impact on spotting days; neither of which is accounted for here. Study perspective and time horizon – appropriate. States elsewhere that a three-year horizon will be used in a follow-up analysis. Setting – Strong design Control and intervention – appropriate Cost outcomes – excellent (Please remove line 195 “There were no differences...” as it belongs in results) Statistical analysis – appropriate ***RESULTS*** Is the effect paper reference already available? (You mention it in line 195 but I didn’t see an actual reference) 285: Please include the difference in spotting days without the amenorrheic women. As mentioned in the intro section, including amenorrheic women in the QALYs is only meaningful if there are other mechanisms by which double-uterine closure is superior. Otherwise, all those amenorrheic women are just adding random variation and may skew the results towards the null. Table 1: If available, it would very much strengthen the paper to know what modes of feeding and contraception were used. And to discuss/consider these in the results. Other things that might have been valuable include question about previous uterine bleeding prior to pregnancy and history of abdominal surgery. Also, is there a reason there’s no “difference” column? Table 2: Please redefine all the costs here, particularly primary and secondary costs, since tables should be able to stand alone from the text. Figure 2: Title needs to add ... “comparing double- to single-layer uterine closure”
--	---

REVIEWER	Alessandra Fenizia George Washington University, USA
REVIEW RETURNED	03-Feb-2021

GENERAL COMMENTS	This paper studies whether double-layer uterine closure is effective at reducing spotting days and cut down costs relative to single-layer uterine closure. The paper is clear, well executed, and well written. I appreciate the effort that the authors put into motivating their work and why it is important to conduct an economic analysis alongside clinical trials. I suggest adding a column to Table 1 containing the mean difference and the 95%
---

	confidence interval. These statistics should be computed using the same methodology used for Table 2 to make these tables directly comparable. Ideally, the authors want to show that the two groups are statistically indistinguishable at baseline. Most baseline characteristics appear fairly balanced but there are a few differences that may turn out to be statistically significant (e.g., smoking habits, hypertension, diabetes mellitus etc.). Does the difference in the cost-effectiveness of these two procedures vary over-time? What matters for policymaking is both the long-run cost-effectiveness and the time profile of when costs are incurred. The authors state that they will do a three-year follow up to evaluate the two procedures under study and I think this key. It may even be appropriate to have another follow-up at, say, 5 years. In the meantime, it would be great if they could exploit the time dimension of their data to evaluate whether these two procedures appear to be equally effective from the very beginning or whether one procedure may look more attractive in the short run but then turn out to be more costly over a year or so. When the authors discuss the generalizability of their findings, they could compare the average characteristics of the women in their sample to the average woman in the Netherlands or the average woman who receives a c-section. This may enrich their discussion of what populations they would/would not feel comfortable generalizing their findings to.
--	--

VERSION 1 – AUTHOR RESPONSE

Response to the reviewers' comments

REVIEWER #1:

Referee's reaction: This is an economic evaluation alongside a superiority trial comparing double-layer to single-layer uterine closure during a primary CS. The study is well designed and executed. The CEA is performed to highest current standards. The justification for choice of outcome, as well as discussion of mechanisms by which the intervention is proposed to improve outcomes, needs to be strengthened. This could impact how the results are presented.

Comments - introduction:

1. Referee's comment: 62: consider also morbidity from cesarean scar pregnancies

Authors' reaction: We included this rare but potentially severe complication as morbidity in a subsequent pregnancy.

2. Referee's comment: 65,66: both statements need citations

Authors' reaction: Unfortunately we forgot to include references here. We included three references for these statements in the revised version of the paper:

4. Wang CB, Chiu WW, Lee CY, et al. Cesarean scar defect: correlation between Cesarean section number, defect size, clinical symptoms and uterine position. *Ultrasound Obstet Gynecol* 2009;34(1):85-9. doi: 10.1002/uog.6405 [published Online First: 2009/07/01]
5. Bij de Vaate AJ, Brolmann HA, van der Voet LF, et al. Ultrasound evaluation of the Cesarean scar: relation between a niche and postmenstrual spotting. *Ultrasound Obstet Gynecol* 2011;37(1):93-9. doi: 10.1002/uog.8864 [published Online First: 2010/10/30]
6. van der Voet LF, Bij de Vaate AM, Veersema S, et al. Long-term complications of caesarean section. The niche in the scar: a prospective cohort study on niche prevalence and its relation to abnormal uterine bleeding. *BJOG* 2014;121(2):236-44. doi: 10.1111/1471-0528.12542 [published Online First: 2014/01/01]

3. Referee's comment: 67: over what timeline do 30% of women with a niche develop AUB? Please make a stronger argument for why spotting days in the first nine months is an important outcome. Additionally, is reducing spotting days the only mechanism by which you hypothesize a double-layer closure is superior to a single-layer closure in the first nine months?

If so, that changes a great deal about how your analysis should be conducted, because you should leave out all the amenorrhoeic women from your analyses. Leaving them in would only add random variation to outcome and cost estimates.

Please consider grammatical review.

Authors' reaction:

First, regarding the first part of reviewer 1's question: 30% of women develop AUB over a timeline of 6-12 months after caesarean section, as shown in previous prospective studies. (Bij de Vaate et al, UOG 2011, vd Voet et al, BJOG 2014) Since spotting days is strongly associated with niche presence which in turn may be influenced by uterine closure technique, we decided to use this as the primary short-term outcome for this cost-effectiveness study. However, in the long term there may be other outcomes that may be positively affected by double-layer closure. Therefore, we plan to conduct a CEA on the long-term (3 years of follow-up) in the future.

Second, we agree with the approach proposed by the reviewer about conducting the analysis, and actually this is the approach we took in the main analyses. Nonetheless, we agree with the reviewer that this information was not clearly stated in the Methods section which may explain reviewer 1's question. To make it clear, additional information was added to the Methods section, statistical analysis paragraph, as below (line 214-215):

"The main analyses included all participants with a menstrual cycle (i.e., participants with amenorrhoea for any reason were excluded) at 9 months follow-up."

Comments - methods:

4. Referee's comment: In the introduction, you state that sonographically visible indentation is visible at site of previous uterine incision, and that this niche is associated with AUB. You never make the connection that double versus single uterine closure is associated with lower risk of a niche. That seems to be an important mediator of uterine bleeding. Is this investigated in the trial?

Authors' reaction: In line 77-79, we state that double-layer closure is suggested to lead to better scar healing and lower niche prevalence. It might well be that niche prevalence or thin residual myometrium thickness overlying the niche is a mediator in developing AUB, but we did not investigate that in the current trial.

Introduction, line 77-79:

"Nevertheless, previous studies also suggested that double-layer closure may result in better uterine scar healing and lower prevalence of large niches thereby possibly leading to lower medical costs than single-layer closure."^{13 15}

5. Referee's comment: Health outcomes – To my understanding, spotting or irregular menstruation during the postpartum return of menses is a common phenomenon regardless of mode of delivery. Women may expect some measure of menstrual irregularity and may not see a physician for this.

Additionally, their mode of child feeding (exclusive breast, formula, or mixture of both) and postpartum contraceptive method (IUD, Implant, OCPs, Depo) all could have a large impact on spotting days; neither of which is accounted for here.

Authors' reaction: We agree with the reviewer that both the method of child feeding and contraceptive method may have a large impact on the results. To verify whether results would change after adjustment for these potential confounders, we performed an additional sensitivity analysis (SA4) in which we adjust for the type of contraception and breastfeeding at 3- and 9-month follow-up. As shown in table below, the difference between groups regarding the primary outcome spotting days did not change as compared to the main analysis and the other sensitivity analyses.

Table 3: Results of the cost-effectiveness analysis

Effect outcome *	Cost difference, € (95% CI)	Effect difference (95% CI)	ICER €/ effect gained	Distribution of the cost-effectiveness plane			
				North-East	South-East	South-West	North-West
Main analysis – Societal perspective							
Spotting days	292 (-518 to 1096)	0.045 (-0.274 to 0.364)	6480	45%	16%	9%	30%
QALY	292 (-518 to 1096)	-0.004 (-0.013 to 0.005)	-73632	12%	8%	17%	63%
Main analysis – Healthcare perspective							
Spotting days	-32 (-390 to 150)	0.045 (-0.274 to 0.364)	-704	29%	32%	19%	20%
QALY	-32 (-390 to 150)	-0.004 (-0.013 to 0.005)	8004	9%	11%	39%	41%
SA1 – including all women randomised - Societal perspective							
QALY	147 (-729 to 872)	-0.005 (-0.013 to 0.003)	-29192	5%	6%	30%	59%
SA1 – including all women randomised - Healthcare perspective							
QALY	-240 (-1237 to 87)	-0.004 (-0.013 to 0.004)	55741	3%	12%	65%	20%
SA2 – complete case analysis – Societal perspective							
Spotting days	380 (-451 to 1273)	0.149 (-0.452 to 0.138)	2554	74%	13%	2%	11%
QALY	397 (-427 to 1273)	-0.006 (-0.017 to 0.005)	-64515	22%	4%	12%	62%

	1275)	0.003)					
SA2 - complete case analysis – Healthcare perspective							
Spotting days	-9 (-251 to 174)	0.186 (-0.486 to 0.103)	-46	54%	37%	4%	5%
QALY	1 (-255 to 177)	-0.007 (-0.016 to 0.002)	-80	23%	4%	34%	39%
SA3 – per protocol analysis - Societal perspective							
Spotting days	59 (-750 to 859)	0.033 (-0.288 to 0.354)	1774	31%	27%	18%	24%
QALY	59 (-750 to 859)	0.003 (-0.006 to 0.012)	19427	46%	28%	16%	10%
SA3 – per protocol analysis – Healthcare perspective							
Spotting days	-50 (-403 to 132)	0.033 (-0.288 to 0.354)	-1515	24%	34%	23%	19%
QALY	-50 (-403 to 132)	0.003 (-0.006 to 0.012)	-16593	34%	40%	16%	10%
SA4 – Main analysis adjusted – Societal Perspective							
Spotting days	Spotting days	Spotting days	Spotting days	Spotting days	Spotting days	Spotting days	Spotting days
QALY	QALY	QALY	QALY	QALY	QALY	QALY	QALY
SA4 – Main analysis adjusted – Healthcare perspective							
Spotting days	Spotting days	Spotting days	Spotting days	Spotting days	Spotting days	Spotting days	Spotting days
QALY	QALY	QALY	QALY	QALY	QALY	QALY	QALY

Data are mean (95% CI). *The effect outcome “Spotting days” was multiplied by -1 in the cost-effectiveness analysis to keep the CE-plane interpretable. CI=confidence interval, ICER=incremental cost-effectiveness ratio, QALY=quality-adjusted life-year, SA=sensitivity analysis.

Main analysis: CEA from a societal and a healthcare perspective for spotting days and QALY including only women without amenorrhoea (total=1620, control n=813, intervention n=807).

SA1: CEA from a societal and a healthcare perspective for QALY, including all women randomised in the study after multiple imputation (i.e., without excluding amenorrhoeic women, n=2292)[Supplementary Table S2].

SA2: CEA from a societal perspective using complete cases for spotting days and total societal costs (total=1065, control n=544, intervention n=521) including only women without amenorrhoea (n=1620).

SA2: CEA from a societal perspective using complete cases for QALY and total societal costs (total=1057, control n=541, intervention n=516) including only women without amenorrhoea (n=1620).

SA2: CEA from a healthcare perspective using complete cases for spotting days and total healthcare costs (total=1315, control n=662, intervention n=653) including only women without amenorrhoea (n=1620).

SA2: CEA from a healthcare perspective using complete cases for QALY and total healthcare costs (total=1310, control n=657, intervention n=653) including only women without amenorrhoea (n=1620).

SA3: per-protocol analysis for spotting days and QALY from a societal perspective (total=1620, control n=828, intervention n=792) including only women without amenorrhoea (n=1620).

SA3: per-protocol analysis for spotting days and QALY from a healthcare perspective (total=1620, control n=828, intervention n=792) including only women without amenorrhoea (n=1620).

SA4: main analysis adjusted for the use of contraception and breastfeeding during follow-up from a societal and a healthcare perspective (total=1620, control n=813, intervention n=807)

6. Referee's comment: Cost outcomes: Please remove line 195 "There were no differences..." as it belongs in results

Authors' reaction: We moved this sentence to the results section, the costs paragraph. We do agree that this is a result and does not belong to the methods section.

Comments - results:

7. Referee's comment: Is the effect paper reference already available? (You mention it in line 195 but I didn't see an actual reference)

Authors' reaction: We added the reference of the effect paper. It was not yet available at the moment of initial submission (august 2020), but it is now.

Line 97-98 – methods section:

The study protocol and the effect paper have been published elsewhere^{17 18}

Line 289-290 – results section – costs paragraph:

There were no differences in days of hospital stay between the groups¹⁸

8. Referee's comment: 285: Please include the difference in spotting days without the amenorrhoeic women. As mentioned in the intro section, including amenorrhoeic women in the QALYs is only meaningful if there are other mechanisms by which double-uterine closure is superior. Otherwise, all those amenorrhoeic women are just adding random variation and may skew the results towards the null.

Authors' reaction: As stated above, the main analysis was performed without amenorrhoeic women. We clarified this issue in the revised manuscript. Only sensitivity analysis 1 was done including the total group, i.e. including also amenorrhoeic women.

We agree with the reviewer that this was not clear in the original manuscript. Therefore, we added a sentence to the analysis on the Statistical Analysis to clarify this, lines 214-215.

"The main analyses included all participants with a menstrual cycle (i.e., participants with amenorrhoea for any reason were excluded) at 9 months follow-up."

9. Referee's comment: Table 1: If available, it would very much strengthen the paper to know what modes of feeding and contraception were used. And to discuss/consider these in the results. Other things that might have been valuable include question about previous uterine bleeding prior to pregnancy and history of abdominal surgery. Also, is there a reason there's no "difference" column?

Authors' reaction: We would like to thank the reviewer for this important comment. We added information on the use of several modes of contraception and child feeding in the Results section of the manuscript (line 277-279). Because this information was collected at 9 months of follow-up, we do not think it is appropriate to include this information in the baseline characteristics table.

Lines 277-279:

"At 9 months follow-up, 12.1% of women in the single-layer arm and 17.1% of women in the double-layer arm were breastfeeding their child(ren). In the single-layer arm, 40.3% of the participants used hormonal contraceptives at 9 months follow-up, and in the double-layer arm this was 38.2%."

Moreover, these covariates were included in sensitivity analysis 4 to account for possible confounding. We added a sentence in the Methods section, sensitivity analysis paragraph, lines 253-255:

“Finally, we performed a sensitivity analysis in which we adjusted for hormonal contraception and breastfeeding (exclusively or combined with formula) at 9 months of follow-up (SA4).”

We decided to not add a ‘mean difference’ column in Table 1. The randomisation should ensure that there are no differences between the group. In previously published papers in *BMJ Open*, this is not common either. We based our decision on a paper of de Boer et al., in which they explain why we should not test for baseline differences. (de Boer et al., *Int J Behav Nutr Phys Act* 2015)

Table 1: Baseline characteristics of women assigned to the intervention or control group

	Single-layer (N=813)	Double-layer (N=807)
Age, years	32.1 (4.7)	32.0 (4.6)
Level of education		
Low	50 (6.5)	54 (7.1)
Middle	263 (34.2)	242 (31.8)
High	452 (58.7)	457 (60.1)
Nulliparous women	568 (73.9)	578 (75.9)
BMI kg/m ²	26.4 (4.5)	26.7 (4.9)
Smoking habit	44 (5.7)	37 (4.9)
Hypertension	146 (19.0)	127 (16.7)
Diabetes mellitus	89 (11.6)	66 (8.6)
Gestational age	38.6 (2.4)	38.6 (2.3)
Previous miscarriage	255 (33.2)	221 (29.0)
Previous ectopic pregnancy	10 (1.3)	12 (1.6)
Planned CS	504 (62.0)	503 (62.3)

Data are mean (SD), or n (%). N is equal to the total number of patients in the group. BMI, body mass index. *5.5% missing data for all variables, except ‘planned CS’ (0%). †5.9% missing data for all variables, except ‘planned CS’ (0%).

Unfortunately, we do not exactly know whether women had previous abdominal surgery. However, exclusion criteria for participation in the trial were, amongst other criteria, 1) menstrual disorder (including heavy or irregular menstrual bleeding) and 2) uterine surgery. We believe that the latter is possibly of large influence on menstrual cycle en uterine healing after caesarean, more than abdominal surgery in general.

As indicated under question 5, we performed an additional sensitivity analysis (SA4) to verify whether results would change after adjustment for contraceptive method and breastfeeding. Results of this

sensitivity analysis were comparable to the results of the main analysis. We added this information in Table 3.

10. Referee's comment: Table 2: Please redefine all the costs here, particularly primary and secondary costs, since tables should be able to stand alone from the text.

Authors' reaction: We included a description of the cost categories in the legend of the table, to enable readers to understand the table apart from the text

“Primary care: costs of visits to general practitioners, health professionals, and complementary healthcare providers. *Secondary care:* costs of ambulatory hospital visits, visits to other healthcare organizations and hospital admissions. *Medication costs:* costs of medication use after discharge from the hospital. *Informal care costs:* costs of received care from family and/or friends due to health problems. *Absenteeism costs at paid work:* costs of sickness absenteeism from paid work. *Absenteeism costs at unpaid work:* costs of absenteeism from unpaid work activities (e.g., household tasks, childcare, voluntary work). *Presenteeism costs:* costs of working while suffering from health.

11. Referee's comment: Figure 2: Title needs to add ... “comparing double- to single-layer uterine closure”

Authors' reaction: We changed the title of figure 2 accordingly, thank you for the input.

REVIEWER #2:

Referee's reaction: This paper studies whether double-layer uterine closure is effective at reducing spotting days and cut down costs relative to single-layer uterine closure. The paper is clear, well executed, and well written. I appreciate the effort that the authors put into motivating their work and why it is important to conduct an economic analysis alongside clinical trials.

1. Referee's comment: I suggest adding a column to Table 1 containing the mean difference and the 95% confidence interval. These statistics should be computed using the same methodology used for Table 2 to make these tables directly comparable. Ideally, the authors want to show that the two groups are statistically indistinguishable at baseline. Most baseline characteristics appear fairly balanced but there are a few differences that may turn out to be statistically significant (e.g., smoking habits, hypertension, diabetes mellitus etc.).

Authors' reaction: As indicated under question 9 of reviewer 1, we decided to not add a 'mean difference' column in Table 1. Randomisation should ensure that there are no differences between the group. In previously published papers in BMJ Open, it is not common either to include a 'mean difference' column to the baseline characteristics table. De Boer et al. explain why we should not test for baseline differences in their paper. (de Boer et al., Int J Behav Nutr Phys Act 2015)

2. Referee's comment: Does the difference in the cost-effectiveness of these two procedures vary over-time? What matters for policymaking is both the long-run cost-effectiveness and the time profile of when costs are incurred. The authors state that they will do a three-year follow up to evaluate the two procedures understudy and I think this key. It may even be appropriate to have another follow-up at, say, 5 years. In the meantime, it would be great if they could exploit the time dimension of their data to evaluate whether these two procedures appear to be equally effective from the very beginning or whether one procedure may look more attractive in the short run but then turn out to be more costly over a year or so.

Authors' reaction: We thank the reviewer for input on this matter. When we look at the course of the number of spotting days in the two groups, then this does not differ between groups. We do not expect that these results regarding spotting days will change in the long-term. However, we think it is

essential to evaluate whether double-layer closure will impact clinical long-term outcomes, such as fertility or uterine rupture. When we have these long-term results, we can make definitive recommendations regarding which closure technique is most effective and cost-effective.

As stated in the protocol article (Stegwee et al., *BMC Pregnancy and Childbirth* 2020) a three year follow-up is planned including gynaecological outcomes (menstrual complaints, sexual functioning, QoL), but also reproductive outcomes: we will collect data about next pregnancies and deliveries, including obstetric complications.

3. Referee’s comment: When the authors discuss the generalizability of their findings, they could compare the average characteristics of the women in their sample to the average woman in the Netherlands or the average woman who receives a c-section. This may enrich their discussion of what populations they would/would not feel comfortable generalizing their findings to.

Authors’ reaction: We thank the reviewer for this comment, on which we do agree on. We added a sentence in the discussion section regarding generalizability of the results (lines 380-382):

“Additionally, generalizability may be impaired since in our study sample relatively many planned CS were performed compared to the Dutch average⁴⁷, probably resulting in an overall underestimation of niche related postmenstrual spotting.”

We also added a row in the baseline characteristics table 1 with ‘planned CS’ as variable. This percentage is about 61% in both groups.

VERSION 2 – REVIEW

REVIEWER	Alessandra Fenizia George Washington University, USA
REVIEW RETURNED	26-Mar-2021
GENERAL COMMENTS	The paper claims that “At baseline, no meaningful differences were found between both groups (Table 1)” (page 13). The authors cannot make such a claim without performing any statistical test. In the first round of refereeing, I suggested adding a column to Table 1 containing the mean difference and the 95% confidence interval. My comment was not addressed in the revision of the paper. I reiterate that if the authors want to claim that the two groups are not meaningfully different at baseline, they have to perform a test. I appreciate that the authors added a brief discussion comparing their sample to the average Dutch woman when they discuss the generalizability of their findings.